# Predicting 3D structure by latent posterior sampling

## Abstract

The remarkable achievements of both generative models of 2D images and neural field representations for 3D scenes present a compelling opportunity to integrate the strengths of both approaches. In this work, we propose a methodology that combines a NeRF-based representation of 3D scenes with probabilistic modeling and reasoning using diffusion models. We view 3D reconstruction as a perception problem with inherent uncertainty that can thereby benefit from probabilistic inference methods. The core idea is to represent the 3D scene as a stochastic latent variable for which we can learn a prior and use it to perform posterior inference given a set of observations. We formulate posterior sampling using the score-based inference method of diffusion models in conjunction with a likelihood term computed from a reconstruction model that includes volumetric rendering. We train the model using a two-stage process: first we train the reconstruction model while auto-decoding the latent representations for a dataset of 3D scenes, and then we train the prior over the latents using a diffusion model. By using the model to generate samples from the posterior we demonstrate that various 3D reconstruction tasks can be performed, differing by the type of observation used as inputs. We showcase reconstruction from single-view, multi-view, noisy images, sparse pixels, and sparse depth data. These observations vary in the amount of information they provide for the scene and we show that our method can model the varying levels of inherent uncertainty associated with each task. Our experiments illustrate that this approach yields a comprehensive method capable of accurately predicting 3D structure from diverse types of observations.

## 1 Introduction

3D prediction based on neural network representations has been a major focus of research in recent years. This line of research, based on the work of Mildenhall et al. (2020); Sitzmann et al. (2019); Park et al. (2019), focuses on two distinct problems. The first problem is **3D reconstruction**, where a 3D representation is predicted from a given set of images of a scene, possibly containing very few or even only one image. A second problem is **3D generation**, where a generative model is used to generate samples of new 3D scenes using various types of conditioning signals like text descriptions or visual data.

While 3D generation is typically addressed using a probabilistic generative model used to sample the representation of new scenes, the 3D reconstruction problem is usually approached with deterministic gradient based optimization methods. However, 3D reconstruction is in most cases an ill-posed problem that requires incorporating prior knowledge and could therefore benefit from probabilistic inference methods.

In this work we propose to tackle the 3D reconstruction problem using a probabilistic framework. Our goal is to predict the full distribution over the 3D scene structure given observations of various types that can carry different levels of information about the scene. The approach consists of a generative model of the latent 3D representation that is used as a **prior**, and a reconstruction model that is used to compute the **likelihood** term. Given a set of observations, predicting the 3D scene becomes a probabilistic inference problem that involves computing the **posterior** distribution over the latent representation.

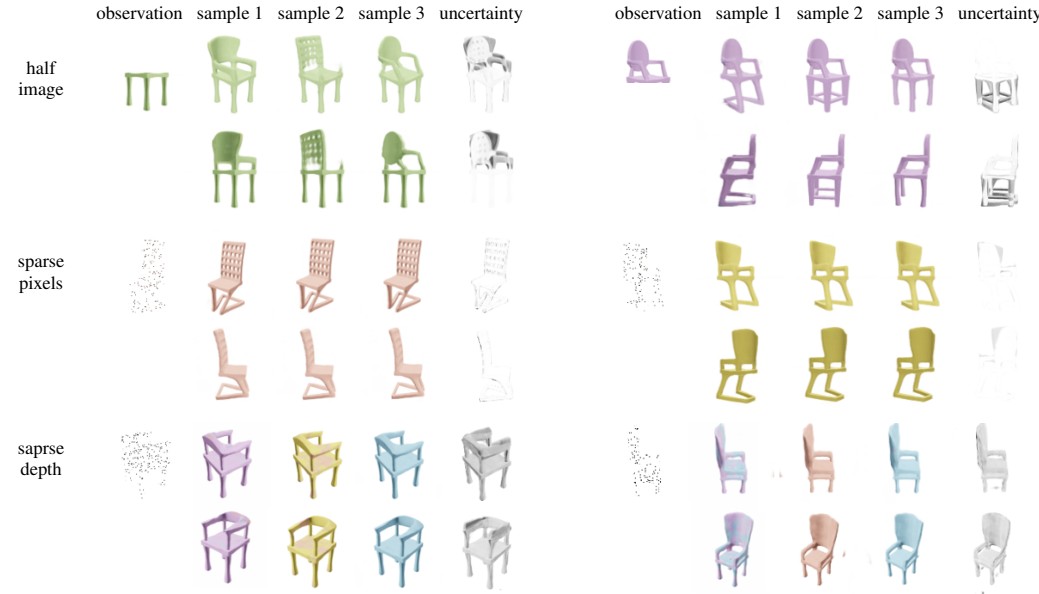

Figure 1: Examples of various 3D prediction tasks performed by generating posterior samples with our method. We train a reconstruction model and a diffusion prior and use them to formulate score-based posterior sampling. For each task we show the observation, three samples of the scene and an uncertainty map computed from the variance of 10 samples. Two different views are shown in subsequent rows. **Top:** reconstruction from half of an image. The variance is high in the hidden half of the scene. **Middle:** reconstruction from only a few pixels (5% of a single image). **Bottom:** reconstruction from a few depth values (5% of a full depth image from a single direction). Samples and uncertainty map suggest sparse depth is enough to reconstruct the 3D shape and uncertainty remains only about color.

To implement this, we follow the work of Dupont et al. (2022) and use a reconstruction model that consists of a volumetric renderer based on a shared **conditional neural field (CNF)** and a latent representation per scene. In order to train the CNF and a prior over the latent representations we use a two-stage training process. In the first stage we use **auto-decoding** to train the shared CNF and optimize the latent representation of each scene in the training set. In the second stage we train a diffusion model over the latent representations of the training set in order to capture the prior distribution.

We use a latent representation that is mapped to the CNF through a tri-plane structure, a representation that provides a tradeoff between global information and local structure and was proven to be effective in previous work (Chan et al., 2021; Chen et al., 2022). As was also shown for 2D data (Bauer et al., 2023), we argue that spatial structure is important for training downstream models such as generative models of the representation.

Training a diffusion model over the latents and combining it with the reconstruction model as a prior, results in a latent variable model that can be used to perform probabilistic reasoning about 3D structure. Here, we propose to use guided iterative Langevin sampling from the diffusion model in order to cover the posterior distribution of the latents. This is in contrast to prior work that suggested to amortize posterior inference (Kosiorek et al., 2021a) in a variational autencoder setting, which empirically demonstrated only limited results. By guiding the Langevin sampling with the gradient of the reconstruction model we combine the strength of two recently successful methods: (1) iterative sampling with diffusion models and (2) gradient based optimization for translating observations to 3D representations. Our approach unifies both methods leading to efficient and accurate 3D reconstruction from observations.

We demonstrate our method on various tasks, highlighting the benefits of generating multiple samples from the posterior in cases where uncertainty is large. We show that this leads to better coverage of the ground truth structure and allows constructing uncertainty maps of the 3D scene.

Our method achieves good results on well studied tasks like single-view reconstruction, and can also be used for infinitely many other tasks which can be defined at test time by formulating a reconstruction term. We show results for 3D reconstruction from noisy observations, sparse pixels and sparse depth information.

Our contributions can be summarized as:

1. We propose to reconstruct 3D scenes by sampling the posterior of a compressed 3D latent representation, using a pre-trained diffusion prior and a conditional NeRF-based decoder.

2. We develop an efficient two-stage training method that (1) auto-decodes a compressed representation of 3D scenes, and (2) trains a diffusion model as a prior over the representation.

3. We demonstrate the usefulness of this approach on various 3D reconstruction tasks, namely, using sparse views, sparse pixels, noisy images and sparse depth data.

4. We show that considering the full posterior can lead to better reconstruction and provide additional insight such as 3D uncertainty maps.

All our code, models and data will be made available upon publication.

## 2 RELATED WORK

**Latent variable models over 3D scenes** Early work on learning probabilistic latent variable models of 3D scenes like GQN (Eslami et al., 2018), was based on the variational autoencoder and did not include any specialized 3D-geometry-based architectures. While the experiments were performed on data with limited complexity, the results already demonstrated the ability to perform probabilistic reasoning and capture uncertainty in a 3D consistent way. Later attempts to equip such models with a 3D rendering pipeline based on NeRF (Kosiorek et al., 2021b), retained the good probabilistic abilities of the model but did not seem to leverage the capacity of NeRF in modeling complex 3D scenes. Our work replaces the amortized variational inference approach in favor of a high capacity diffusion model and Langevin posterior sampling. More recent works (Shen et al., 2022; Sünderhauf et al., 2022; Goli et al., 2023) target uncertainty modeling using different stochastic and Bayesian methods to detect reconstruction inconsistencies in a single scene, however they do not use a data-driven learned prior like we do.

**Generating 3D with 2D Generative models** The scarcity of ground truth data in 3D has led various work on methods to generate 3D data using generative models of 2D images. Notable works use pretrained image diffusion models in order to train a NeRF representation (Poole et al., 2022; Watson et al., 2022; Liu et al., 2023; Sargent et al., 2023). In this approach text conditional diffusion models can be used to generate 3D from text descriptions, and multiview 2D diffusion models can be used to generate 3D scenes conditioned on a single input image. In a similar approach, Liu et al. (2024) use a 2D prior to compute 3D uncertainty maps. While these models lead to impressive results for 3D generation and clearly demonstrate a level of 3D understanding, they do not explicitly reason about the latent 3D structure of the scene, preventing them from performing the full range of 3D probabilistic reasoning tasks, e.g. reconstruction from depth information.

**Generative models of observed 3D representations** Despite the scarcity of 3D data, some work has studied generative modeling directly on various datasets of 3D representations. Shue et al. (2022) trains a diffusion model on a tri-plane representation. Erkoç et al. (2023) trains a diffusion model on 3D shape data, by first training neural field MLPs and then modeling a prior over them using a transformer based denoiser. Jun & Nichol (2023) uses a dataset of point clouds. It first trains an encoder that maps the point cloud to neural field MLP weights, and then trains a (conditional) diffusion model on the weights. In our work there is no need for ground truth 3D representations as we only rely on 2D image datasets of 3D scenes.

**Generative models of latent 3D representations** Our work is inspired by Dupont et al. (2022), where it is proposed to represent continuous functions such as images and 3D scenes as conditional neural fields (CNF) by first training a CNF, and then using the auto-decoded conditioning vector as data for training downstream models such as generative models. Their paper provides a simple proof of concept of using such a generative model as a prior and perform MAP inference. Compared to their models and experiments on 3D data, we introduce the tri-plane representation which

results in a better and more compressed latent representation, we circumvent the expensive meta-learning process, and we demonstrate better results by using a diffusion model to sample from the full distribution of the posterior.

Another work that provides inspiration to our model is Bautista et al. (2022). As far as we can tell our model shares a similar structure to theirs (their code is not open sourced), however they focus on the task of 3D generation rather than 3D reconstruction, and show results for unconditional generation and separately-trained conditional generation based on text description and image data. Yang et al. (2023) and Chen et al. (2023) also train diffusion models using a tri-plane representation, however they do it directly on the tri-planes rather than using a compressed latent representation as we do, and focus on 3D generation and single/multi-view 3D reconstruction. The core idea of Chen et al. (2023) is the unification of the two stages of training into one combined training process which is orthogonal to our contributions and could be used for our method as well. The concurrent work of Le et al. (2024) shares a similar motivation to ours. They focus on specific type of noisy observations using a 3D modeling of the corruption field, and they extensively demonstrate the advantages of the full posterior distribution over the maximum only (MAP inference). In Zhang et al. (2024) a 3D generative model is trained based on a Gaussian splatting representation which could also be combined with posterior sampling in future research.

# 3 BACKGROUND

## 3.1 AUTO-DECODING 3D REPRESENTATIONS

Recent years have witnessed breakthroughs in the representation of 3D scenes with deep neural networks. Most notable is NeRF (Mildenhall et al., 2020) - a model that employs a neural network to capture a 3D scene by training it with a reconstruction loss based on rendering various views of the scene. While in NeRF, a different model needs to be trained for each 3D scene, follow-up work has focused on developing a generalizable model that can leverage prior information of 3D scenes, and can consequently be trained with fewer views per scene. Earlier models like PixelNeRF (Yu et al., 2021) and IBRNet (Wang et al., 2021) apply encoder networks in order to learn image features, which are then projected into the 3D scene. In contrast to vanilla NeRF models that consist of a neural field (NF) network, generlizable NeRF models often rely on a *conditional* neural field (CNF) where the NF network is shared across scenes, but conditioned on different representations per scene.

More recently, different papers propose to use CNFs to train representations of scenes that can later be used in downstream tasks (Dupont et al., 2022; Bautista et al., 2022; Chen et al., 2023; Yang et al., 2023). Rather than training an encoder to map images to a scene representation, these models use an *auto-decoding* approach (Bojanowski et al., 2019; Park et al., 2019), where the representations are optimized for each scene concurrently with the training of the shared CNF. Most of these models are based on a *tri-plane* representation (Chan et al., 2021; Chen et al., 2022). This representation is structured as three planes posed in three orthogonal directions in the scene (Fig. 2). The representation is used as a conditioning input to the CNF by projecting the queried 3D position into the different planes, interpolating the values for each plane, and concatenating the results. This representation has proven very effective in 3D modeling, as it maintains a spatial structure in 3D space, and balances between global and local information.

## 3.2 POSTERIOR SAMPLING WITH DIFFUSION MODELS

Diffusion models, particularly Denoising Diffusion Probabilistic Models (DDPM) (Ho et al., 2020), have gained significant attention in the machine learning community for their ability to generate high-quality samples from complex data distributions. Many different variants stemming from different formulations have been developed in recent years (Sohl-Dickstein et al., 2015; Song & Ermon, 2019; Ho et al., 2020; Song et al., 2020). The model is trained to invert a forward diffusion process $x_t = \sqrt{\alpha_t} x_{t-1} + \sqrt{1 - \alpha_t} \epsilon$ where the noise at each step is defined as $\epsilon \sim \mathcal{N}(0, I)$ and $\alpha_t$ are parameters that determine the schedule of the noise level. Most of the models are based on a *U-net* architecture (Ronneberger et al., 2015), that predicts the total additive noise at time step $t$, $\epsilon_\theta(x_t, t)$.

By predicting the noise, a sample of $x_{t-1}$ and an estimate of the clean image $\hat{x}_0$ are computed using:

$$x_{t-1} \sim \mathcal{N}\left(\frac{1}{\sqrt{\alpha_t}}\left(x_t - \frac{1-\alpha_t}{\sqrt{1-\bar{\alpha}_t}}\epsilon_\theta(x_t, t)\right), \tilde{\beta}_t I\right) \qquad (1)$$

$$\hat{x}_0(x_t, t) = \frac{1}{\sqrt{\bar{\alpha}_t}}\left(x_t - \sqrt{1-\bar{\alpha}_t}\epsilon_\theta(x_t, t)\right) \qquad (2)$$

where $\bar{\alpha}_t = \prod_{s=0}^{t} \alpha_s$ and $\tilde{\beta}_t$ is a fixed variance usually set to $\tilde{\beta}_t = \frac{(1-\bar{\alpha}_{t-1})(1-\alpha_t)}{1-\bar{\alpha}_t}$.

After training, the model can be used to generate samples by iteratively applying Eq. 1 starting from Gaussian noise. This process can also be viewed as a Langevin sampling process, where the estimated noise at each step $\epsilon_\theta(x_t, t)$ is considered an approximate score function $\nabla \log p_t(x_t)$.

Following the development of these models, many work propose to use diffusion models as image priors, and formulate different image restoration tasks like denoising, inpatining, and deblurring as posterior sampling (Choi et al., 2021; Chung et al., 2022a;b; Chung & Ye, 2022; Graikos et al., 2022; Song et al., 2021; Feng et al., 2023; Chung et al., 2023; Jalal et al., 2021; Kawar et al., 2022; Song et al., 2023; Bansal et al., 2023; Bar Nathan et al., 2024). Under the Langevin sampling view, the iterative sampling procedure that is usually used to generate samples from the prior is extended to include a likelihood term, which is formulated from a forward model that maps the image to observation space, where a loss function is applied.

While sampling from the prior involves computing the prior score function $\nabla_{x_t} \log p_t(x_t)$ based on the output of the U-net at each step $t$, sampling from the posterior is implemented by adding the gradient of the likelihood term. This method of adding additional terms in each step of the inverse sampling iterations is also known as *guidance*. More formally, given an observation $y$ and a likelihood function $p(y|x)$, posterior sampling is computed using the posterior score, defined as:

$$\nabla_{x_t} \log p_t(x_t|y) = \nabla_{x_t} \log p_t(x_t) + \nabla_{x_t} \log p_t(y|x_t) \qquad (3)$$

Since the exact likelihood model is usually based on the clean image $x_0$ rather than a noisy image $x_t$, different approximation to this have been proposed. We describe our approximation method in detail in section 4.3.

## 4 METHOD

In this section we describe our method both at training time and at inference time. Training is based on two stages: (1) training the reconstruction model while optimizing the latent representation of the training scenes (auto-decoding), and (2) training a diffusion model over the latents as a prior. At inference time we use the trained prior and reconstruction models to perform posterior sampling of the latents. For all implementation details please refer to Sec. A in the appendix.

### 4.1 TRAINING THE REPRESENTATION AND RECONSTRUCTION

The reconstruction model is a CNF followed by a volumetric renderer. Conditioned on a scene representation, the CNF predicts the values of 3D positions within the scene that are subsequently used by the volumetric renderer. The CNF is trained while concurrently auto-decoding the representation of each scene. The role of the reconstruction model is to form a mapping from the representation vectors to the values of the observations, i.e. image pixels, and also serve as the model through which the representation is optimized, effectively mapping the 3D scene observations back into the representation.

The model is depicted in Fig. 2. The latent vector $z_i \in \mathbb{R}^d$, corresponding to the $i$-th scene, is first reshaped into a 2D map of shape $r \times r \times c$. The latent decoder $D1$ decodes $z_i$ into a 3D tensor $T \in \mathbb{R}^{R \times R \times 3C}$ using a series of ResNet blocks. $T$ is reshaped to form a tri-plane representation $T_{1i}, T_{2i}, T_{3i} \in \mathbb{R}^{R \times R \times C}$. The tri-planes structure is used for reconstruction as follows: given an image of a scene, rays are projected from each pixel into the 3D scene, and multiple 3D points are sampled along each ray. Each 3D point is projected onto the tri-planes, and using bi-linear interpolation, each plane produces a single corresponding feature vector $f \in \mathbb{R}^C$. The three feature vectors are concatenated to form $f^*$, which is used as the input to the decoder $D2$, an MLP that transforms the tri-plane interpolated vector into $RGB$ and $\sigma$ values for the corresponding 3D position.

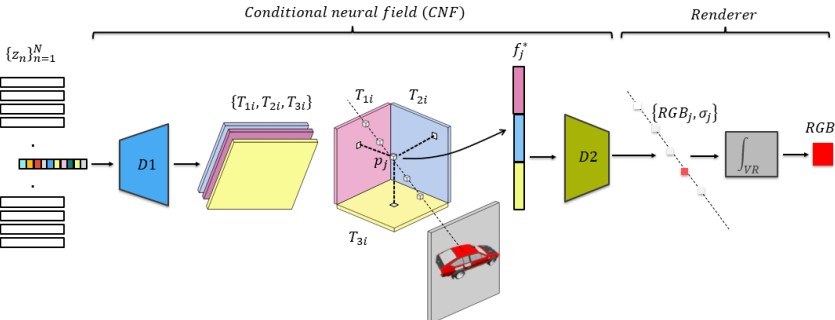

Figure 2: The reconstruction model mapping latent representations to images of a 3D scene. The latent decoder $D1$ maps the latent vector $z_i$ corresponding to scene $i$ into three multichannel planes (tri-planes) $\{T_{1i}, T_{2i}, T_{3i}\}$. Given an image and a camera position from which the image was taken, a ray is projected onto the scene from each pixel of the image, and multiple 3D points are sampled along the ray. Each 3D point $p_j$, is projected onto the multichannel tri-planes where each plane produces a feature vector $f_j$ using bilinear interpolation. The three feature vectors are concatenated to form one feature vector $f_j^*$, and the decoder $D2$ is used to produce RGB and $\sigma$ values for each 3D point along the ray. Volumetric rendering is then used to generate a single RGB value to be compared to the ground truth value of the pixel in the image.

This process is repeated for all 3D points along the ray, and then volumetric rendering is applied to generate a single RGB value for the pixel from which the ray was projected into the scene.

The reconstruction model (RM) and latent representations are trained using the auto-decoding approach as following: first all network weights $\phi$ are randomly initialized, and a latent vector $z_i$ initialized to zero is assigned to each scene $i$ in the dataset. Then, at each training iteration, a minibatch of scenes $\mathcal{B}$ is randomly selected along with the corresponding latent vectors, where for each scene a random set of images, and random set of pixels within the images are used. The minibatch is used to apply a forward pass of the reconstruction model on the latents, and backpropagate the loss between the model's output and ground-truth pixel values to all network weights and latent values.

$$\mathcal{L}_{rec} = \sum_{i \in \mathcal{B}} \sum_{x \in \mathcal{X}_i} \|x - RM_\phi(z_i)\|^2 \tag{4}$$

where $\mathcal{B}$ is a random minibatch of scenes, and $\mathcal{X}_i$ is a random set of pixels from a random set of images from each scene $i$. The network weights are updated using $\partial \mathcal{L}_{rec} / \partial \phi$, and the latents are updated using $\partial \mathcal{L}_{rec} / \partial z_i$. In this way the latent representation for each scene is optimized while the network weights converge to their final values. For all experiments in the paper we use a latent dimension of 1024, which forms a highly compressed representation of the scenes. For more implementation details see Sec. A.

Fig. 3 shows examples of reconstruction for a few selected scenes using two models that were trained on the SRN cars and Objaverse chairs datasets (see Sec. 5 for details about the datasets). After training the reconstruction models, 125 images of held-out test scenes are used to optimize the scene latents while freezing the reconstruction model's weights, and the latents are then used to reconstruct novel views of the scenes. The results show that the latent representation captures the 3D scenes with high fidelity. In Tab. 1 we compare the reconstruction accuracy of our compressed representation to Dupont et al. (2022). Our results are favorable, and we argue that this is due to the spatial structure of the tri-plane representation.

### 4.2 TRAINING THE PRIOR

The goal of the second stage is to obtain a prior over the latent representation. This is achieved by training a generative model based on a Denoising Diffusion Probabilistic Model (DDPM) (Ho et al., 2020) on the latent data obtained in the first stage. As is standard in diffusion models, the model is based on a U-net architecture (Ronneberger et al., 2015) that is trained to denoise the latent representations $\{z_i\}_{n=1}^N \in \mathbb{R}^d$. To comply with the U-net architecture, the latents are reshaped to be

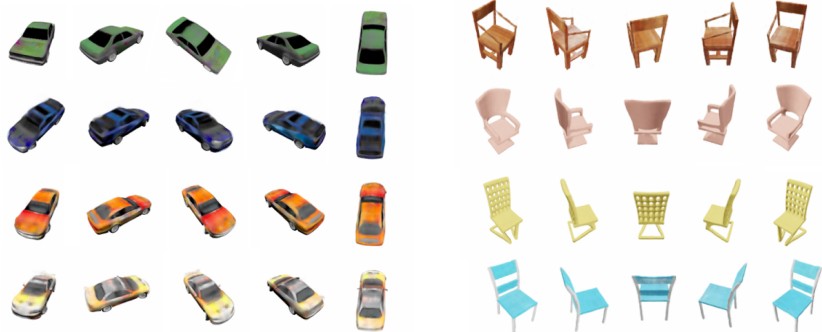

| | functa, as reported in (Dupont et al., 2022) | ours |
|---|---|---|
| latent dim | 1024 | 1024 |
| PSNR train | 24.4 | 27.67 |
| PSNR test | 23.1 | 26.9 |

Table 1: Reconstruction from latent.

Figure 3: Novel view reconstruction for held out 3D scenes. Each pair shows the ground truth image (left) and the reconstructed image (right). Top row: SRN cars. Bottom row: Objaverse chairs.

Figure 4: Samples from the trained diffusion model. Each row corresponds to a different sample of the latent representation, corresponding to a different 3D scene, and each column shows a different view reconstructed from the same scene. Left: SRN cars. Right: Objaverse-lvis chairs.

$\{z_i\}_{n=1}^{N} \in \mathbb{R}^{r \times r \times c}$. The training loss is computed by:

$$\mathcal{L}_{gen} = \mathbb{E}_{z \in \{z\}, \epsilon \in \mathcal{N}(0,1), t \in U[0,T]} \left\| \epsilon_\theta \left( \sqrt{\bar{\alpha}_t} z + \sqrt{1 - \bar{\alpha}_t} \epsilon, t \right) - \epsilon \right\|^2 \tag{5}$$

Our implementation is based on Graikos et al. (2022) (more details in Sec. A). Fig 4 shows examples of random samples generated from the learned prior. Each image is generated by first sampling a latent from the prior, corresponding to a sampled scene (rows), and then using the reconstruction model to render images of the scene from different views (columns). The resulting samples show both coherence and diversity.

### 4.3 SAMPLING FROM THE POSTERIOR

As described in Sec. 3.2, different methods have been proposed to sample from posterior distributions given a trained diffusion model as a prior. These methods consist of adding a likelihood term to each step in the iterative process of sampling from the prior. Here, the likelihood term comes from applying the reconstruction model (RM) on the estimated latent, and computing a squared loss compared to the given observation $y$, which corresponds to a Gaussian log-likelihood.

$$\log p(y|z) = -s\|y - RM_\phi(z)\|^2 + const. = -\mathcal{L}_{rec} + const. \tag{6}$$

where $s$ is a scaling factor corresponding to the assumed variance of the reconstruction.

The method is depicted in Fig. 5, and described in Alg. 1. In more detail, at each step $t$ the output of the U-net $\epsilon_\theta(z_t, t)$ is used to compute the one-step denoised latent $z_{t-1}$ and a fully denoised estimate $\hat{z_0}$ (Eq. 1 and Eq. 2 respectively). The clean estimate is fed to the reconstruction model (RM) which outputs a prediction of the input views. A gradient of the log-likelihood with repsect to $z_t$ can be computed by back-propagating the reconstruction error (Eq. 6) between the predicted images and the observed ground-truth images, however this requires back-propagating through the U-net at each step. In order to accelerate inference, we approximate this gradient by computing $\tilde{z}_0(z_{t-1}) = \frac{1}{\sqrt{\bar{\alpha}_t}} \left( z_{t-1} - \sqrt{1 - \bar{\alpha}_t} \epsilon_\theta(z_t, t) \right)$, and the gradient with respect to $z_{t-1}$. When using many sampling steps we empirically observe that the difference between $z_t$ and $z_{t-1}$ is negligible and this approximation can be used to efficiently compute the posterior score:

$$\nabla_{z_t} \log p_t(z_t \mid y) \approx \nabla_{z_t} \log p_t(z_t) + \nabla_{z_{t-1}} \log p\left(y \mid \tilde{z}_0(z_{t-1}, t)\right), \tag{7}$$

**Algorithm 1:** Posterior Sampling

**Input:** images $y$, scale $s$
Initialize $z_T \sim N(0, 1)$
**for** $t = T$ **to** $1$ **do**
$\quad \epsilon \leftarrow \text{U-net}_\theta(z_t, t)$
$\quad z_{t-1} \sim \mathcal{N}\left(\frac{1}{\sqrt{\alpha_t}}\left(z_t - \frac{1-\alpha_t}{\sqrt{1-\bar{\alpha}_t}}\epsilon\right), \tilde{\beta}\right)$
$\quad \tilde{z}_0(z_{t-1}) = \frac{1}{\sqrt{\bar{\alpha}_t}}\left(z_{t-1} - \sqrt{1-\bar{\alpha}_t}\epsilon\right)$
$\quad \mathcal{L}_{rec} = \|y - RM_\phi\left(\tilde{z}_0(z_{t-1})\right)\|^2$
$\quad z_{t-1} \leftarrow z_{t-1} - s\partial\mathcal{L}_{rec}/\partial z_{t-1}$

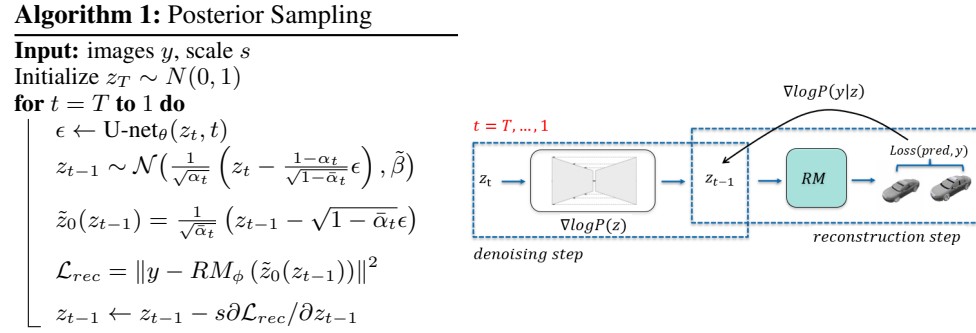

Figure 5: **Left:** The posterior sampling algorithm. **Right:** Illustration of a single step in the iterative process. Conditioned on the previous estimate $z_t$, the U-net predicts the noise, which is used to compute both $z_{t-1}$ and $\tilde{z}_0$. The latter is fed to the reconstruction model to predict an image from the given view which is compared to the ground truth image $y$. The error is backpropagated through the frozen networks to compute a gradient which is then added to $z_{t-1}$.

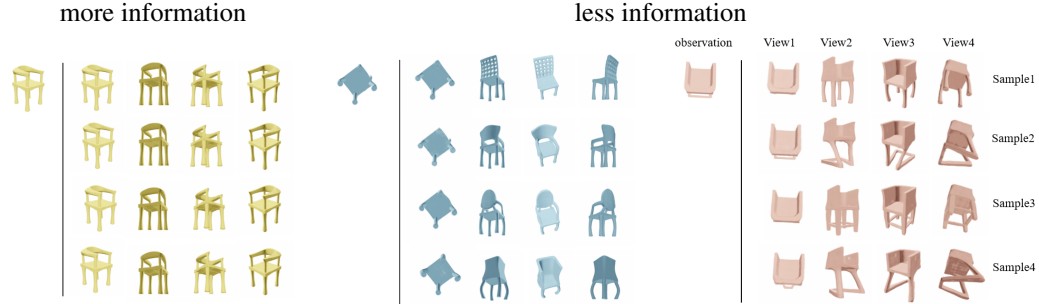

Figure 6: Posterior samples given a single view for Objaverse chairs. Each row corresponds to a different sample of the scene, and each column shows a different view. The observation in the example on the left carries high information about the scene, resulting in very similar samples. The observations in the middle and right scenes are less informative, and therefore result in more diverse samples, where the chairs are completed with different possible configurations of legs, armrests and backrests. These example demonstrate a coherent merging of observed data and prior information.

Repeating this process for $t = T...1$ forms an approximated Langevin sampling process from the posterior distribution.

As the reconstruction loss is calculated with no regards to pixel order or quantity, this approach allows training a single prior model, and then use it to generate posterior samples for various types of conditioning signals. Examples include conditioning on many images, few images, or even a few random pixels per scene. Moreover, the desired inference task does not even need to be known at training time, as long as a corresponding reconstruction term can be formulated and differentiated at inference time.

## 5 EXPERIMENTS

For all experiments we use the same model and the same configuration. See Sec. A for more details.

**Data.** We use two datasets in our experiments. The first dataset is SRN Cars (Sitzmann et al., 2019), which comprises 3,200 scenes with 250 images each. We randomly divide the images in each scene evenly between training images and test images, and we use 3,000 scenes for training, holding out 200 scenes for testing. The second dataset we use is the Objaverse-lvis chair category (Deitke et al., 2022), which comprises 439 instances with 100 images generated for each scene. While this dataset is smaller, it is more diverse in terms of shapes. We use 80% of the images in each scene for training,

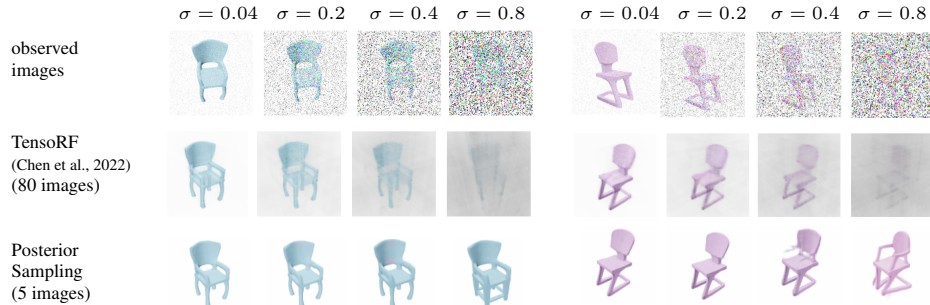

| | | 1-view | | 2-view | |
|---|---|---|---|---|---|
| | | PSNR↑ | SSIM↑ | PSNR↑ | SSIM↑ |
| Point-estimate/hybrid methods | | | | | |
| PixelNeRF(Yu et al., 2021)* | | 23.17 | 0.90 | 25.66 | 0.94 |
| SSDNeRF(Chen et al., 2023) | | 24.49 | 0.92 | 26.77 | 0.95 |
| Multi-sample estimate (ours) | | | | | |
| 1 sample | | 22.78 | 0.87 | 25.71 | 0.91 |
| 5 samples | | 23.38 | 0.88 | 26.26 | 0.92 |
| 10 samples | | 23.48 | 0.88 | 26.35 | 0.92 |
| 20 samples | | 23.55 | 0.88 | 26.40 | 0.92 |

Table 2: Reconstruction given one and two views from held-out SRN cars scenes. *results as reported in the PixelNeRF paper.

Figure 7: Evaluating reconstruction metrics of using multiple samples vs. using a point estimate. Tab. 2 shows that while averaging the images of more scene samples leads to better performance, it still doesn't outperform well-tuned point-estimate or hybrid methods. However, samples on the left suggest that given a less informative view, point-estimate and hybrid methods generate samples that "average out" the uncertainty and fail to capture diverse plausible reconstructions like our method.

Figure 8: 3D reconstruction from noisy images. Reconstruction from 80 images without a prior (TensoRF) quickly deteriorates as noise increases. Using our prior to perform posterior sampling results in a much more robust method, significantly outperforming TensoRF even when using an order of magnitude less images (5 images).

and hold out 8 scenes for testing and visualizations. For both datasets we use image resolution of $128 \times 128$.

We show results of generating posterior samples given one observed image per scene. In Fig. 6, three examples from Objaverse chairs are shown. In the scene shown on the left, the given image contains enough information to predict any view of the scene with certainty. This results in multiple samples (rows) that are almost identical. In the other examples the observed image is less informative and does not provide enough information about the scene from all angles. Therefore, samples from the posterior exhibit more diversity in the way they complete the missing information. More concretely, the chairs observed from uninformative views are predicted to have different possible leg, armrest and backrest configurations. Note that while the samples are different, the generated latent is a 3D representation, so each sample can be used to predict a coherent set of images from different views.

In Fig. 1, we demonstrate the ability of the method to perform more diverse probabilistic reasoning tasks. We show prediction from half-image inputs, from a sparse set of pixels of one image (5%), and from a sparse set of depth map pixels (5% of a depth map from a single view). For each scene we show three samples, showing two different views for each, and an uncertainty map. The uncertainty is computed by generating 10 samples of the scene, rendering corresponding 10 images for each view and computing the variance in the rendered images. Using our method for partial RGB observations (half-image or sparse pixels) is trivial to implement since the reconstruction model operates per pixel and can be used to predict any subset of pixels in the scene. In the case of depth data, we implement a different reconstruction loss comparing the predicted $\sigma$ values to ground truth values without using

the $RGB$ prediction and the renderer in Fig. 2. Given a depth pixel value, the ground truth value of $\sigma$ is set to 1 for the 3D point on the ray sampled at the given depth value, and 0 for all the other 3D points. We emphasize that this reconstruction model is formulated at inference time and is not used at training. The results show the different plausible predictions of the scene and the resulting uncertainty. For the first case we see that the uncertainty is high for the hidden half of the scene as expected. For the other two cases, samples generated from sparse pixel observations demonstrate a high degree of similarity, suggesting, perhaps surprisingly, that even just 5% of the pixels from a single view is sufficient for accurate 3D scene prediction. In case of the sparse depth data, the only uncertain aspect is the object color.

In Fig. 7 and Tab. 2 we evaluate our method for the well studied tasks of 3D reconstruction given one view and two views, and compare it with PixelNeRF (Yu et al., 2021) and SSDNeRF (Chen et al., 2023). PixelNeRF is a deterministic method that predicts a single point-estimate for each scene, and is specifically designed for conditioning on one or two views. SSDNeRF shares a similar model to ours, using a diffusion-model-based prior and a tri-plane representation. The representation for each scene is the tri-plane structure itself which makes it more than two orders of magnitude larger than our 1024-dimensional representation. While the sampling method is stochastic as ours, it also includes additional regularizations and deterministic finetuning, and we therefore call it a *hybrid* method. We evaluate the reconstruction using the same protocol introduced by Yu et al. (2021) on 49 random scenes from our SRN cars test set (see Sec. A for details). We generate multiple samples, and compute the metrics (PSNR and SSIM) by averaging the image reconstruction across the samples. The results in Tab. 2 show that using more samples improves performance, and that the results approach the state-of-the-art results but do not surpass them. We hypothesize that this is because point-estimate methods are tuned to directly predict the posterior mean in order to increase metric performance. However, as shown in Fig. 7 these methods fail to predict diverse reconstructions in cases where uncertainty is high. The figure shows that given a non-informative view like the bottom of a car, samples from the hybrid method of SSDNeRF are all similar and of average color, while our method produces a highly diverse set of samples. This also means that these point-estimate or hybrid methods cannot provide reliable measures of uncertainty like we show above for our method.

Finally, we show results for 3D reconstruction from noisy images of a scene and demonstrate the robustness of our method. We generate samples given multiple images of a scene and compare our method to TensoRF (Chen et al., 2022) without a prior, using the code that was used as a basis for our reconstruction model. We repeat the process for increasing noise levels. The results in Fig. 8 show that posterior sampling can lead to significant increase in robustness to noise. While TensoRF is trained with 80 images and suffers from a rapid decline in performance as noise increases, our method, even when using only 5 images from the scene, shows only a mild decrease in performance due to noise and can predict plausible and coherent completions to missing information.

## 6 Conclusion

In this work we introduced a methodology that combines the strengths of NeRF-based 3D reconstruction together with the probabilistic reasoning of diffusion models. Our method views 3D reconstruction as an ill-posed perception problem that requires reconciling the observed information with prior knowledge. We showed that (1) 3D scenes can be efficiently represented by compact latent vectors, using a reconstruction model that consists of a tri-plane representation, which preserves spatial structure within the 3D model; and (2) this representation is amenable to training a strong diffusion-model based prior that can later be used to solve various inference tasks. We highlight the importance of predicting the full posterior distribution rather than relying on point estimates, and demonstrate a comprehensive method that can be used to solve various 3D reconstruction tasks.

**Limitations and future work**: A main challenge that remains in 3D reconstruction is scaling to more complex and more diverse data towards developing methods that can reliably predict real 3D scenes from different levels of observations. Another challenge is the slow sampling time with diffusion models. While our results are demonstrated on small scale data, we believe that the compressed representation and the principled way of handling uncertainty that we propose, combined with recent developments in accelerating diffusion model sampling, are key for scaling up these models to larger and more complex datasets.

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

## A    IMPLEMENTATION DETAILS

In this section we describe the implementation details of our model and experiments. All code, models and data will be made available upon publication.

### TRAINING THE REPRESENTATION AND RECONSTRUCTION MODEL

The dataset size corresponds to the number of latent vectors, with each latent representing a single scene $\{z_i\}_{i=1}^N$ (N scenes = N latents). During training, each scene's images optimize only its respective latent, while the entire model, including latents and decoders, is jointly trained. The latent representation $z_i$ dimensions are $d = 1024$, $r = 16$, $c = 4$. D1 is constructed using a series of six ResNet blocks where at each block the number of channels is the following: [4,32,64,96,128,192]. Blocks are followed by a self-attention layer and alternating upsampling. The resulting 3D tensor $T$ is divided into two tensors responsible for generating RGB and density, $T_{RGB} \in \mathbb{R}^{R \times R \times 3C_{RGB}}$, $T_\sigma \in \mathbb{R}^{R \times R \times 3C_\sigma}$, respectively. $T_{RGB}$ is reshaped to form a tri-plane representation $T_{1i}, T_{2i}, T_{3i} \in \mathbb{R}^{R \times R \times C_{RGB}}$. Similarly, $T_\sigma$ forms a tri-plane representation $T'_{1i}, T'_{2i}, T'_{3i} \in \mathbb{R}^{R \times R \times C_\sigma}$. Dimensions are $R = 128$, $C_{RGB} = 48$, $C_\sigma = 16$. For each scene $i$, we randomly select 4096 rays from pixels in the training images. Along each ray, we sample 220 3D points and project them onto the tri-planes of both the RGB and density planes separately. This projection extracts 3 feature vectors from each of the respective planes for further processing. Three vectors are concatenated into a single feature vector $f^* \in \mathbb{R}^{3C}$ for RGB calculation and $f^* \in \mathbb{R}^C_{RGB/\sigma}$ for density. While the density feature vector $f^*_\sigma$ produces density for 3D points by simply summing its elements, the RGB feature vector $f^*_{RGB}$ is passed through D2 to produce a single RGB value. D2 is an MLP of 7 layers. Once all 3D points along the ray have RGB and density values, volumetric rendering, a parameterless process, produces a single RGB value to be compared with the pixel's color.

We train the model with a minibatch $\mathcal{B}$ size of 2 scenes, and with an Adam optimizer using three different learning rates: 1e-3 for the latents, 1e-4 for the D1 parameters and $1e-3$ for D2 parameters.

Our model is based on the code published in Chen et al. (2022).

At test time, a new latent (initialized to zeros) is coupled with the new scene and optimized using the learned/frozen decoders.

### TRAINING THE PRIOR

As in section 4.1, latent representation $z_i$ dimensions are $d = 1024$, $r = 16$, $c = 4$. The diffusion model used is implemented by Graikos et al. (2022) with the following parameters: The noise scheduler is a linear schedule with parameters $T = 1000$, $\beta_0 = 1e - 4$, $\beta_T = 2e - 2$. The U-net parameters are $model\_channels = 64$, $num\_resnet\_blocks = 2$, $channel\_mult = (1, 2, 3, 4)$, $attention\_resolutions = [8, 4]$, $num\_heads = 4$. We train the model with a minibatch $\mathcal{B}$ size of 32 scenes, and with an Adam optimizer with learning rate equal to 1e-3.

The reconstruction model and the diffusion model were trained on an NVIDIA GeForce RTX 4090 for Approximately one day each.

### EXPERIMENTS

Posterior sampling involves two types of computations: 1) denoising, using the diffusion as a prior to generate a plausible latent, and 2) reconstruction, using the reconstruction model to align the latent with the observed views.

For all experiments we use the same model using the same inference process. We generate posterior samples using 1000 iterations as described in Alg. 1 with the same scale factor $s$ =5e-3 for all experiments. The only exception is the experiment with noisy data 8, where the scale factor for most extreme noise level $\sigma = 0.8$ was decreased to a value of $s$ =3e-3, corresponding to the high noise variance in the observation.

For the evaluations in Tab. 1 and Tab. 2 we use randomly chosen 49 held-out scenes from SRN cars. We use these scenes to evaluate our method and SSDNeRF using their published code.

