# OpenReview forum: "PREDICTING 3D STRUCTURE BY LATENT POSTERIOR SAMPLING"
_ICLR.cc/2025/Conference — Submitted to ICLR 2025_

### Official Review · Reviewer_AMeD · 2024-10-24

**Soundness:** 3
**Presentation:** 3
**Contribution:** 3
**Rating:** 5
**Confidence:** 4

**Summary:**

The paper proposes a probabilistic 3D reconstruction model with diffusion prior. The core idea of this methodology consists of two parts: a two-stage training approach and a posterior inference pipeline. At the first training stage, it uses auto-decoding to train a conditional reconstruction model accompanying with its input latent space on a 3D dataset. Then it trains a diffusion model over the latent. During inference, the model iteratively denoises sampled noise into refined latent for reconstruction, where the reconstruction model provides posterior score at each step. The experiments demonstrate that the proposed framework can be applied on various reconstruction tasks, such as single-view, multi-view, noisy images, sparse pixels, and sparse depth, using some simple datasets. Overall, the paper shows a promising potential in using probabilistic 3D model for reconstruction and possible future development.

**Strengths:**

1. The paper is well-written and easy-to-understand. The background section provides clear preliminary knowledge on the theories used in this paper.
2. The model reasons the 3D structure well. When given less informative inputs, it synthesizes the overall geometry well and generates accurate uncertainty at unknown parts.
3. The model is time-efficient for 3D reconstruction without optimization at inference time.
4. The paper proposes a novel framework to incorporate posterior sampling and 2D diffusion models with 3D reconstruction model for non-deterministic optimization.

**Weaknesses:**

1.	The implementation details are not provided in the main paper, and they are also not complete enough in the appendix. Please provide more details so that the paper can be more reproducible. For example, there are some important hyperparameters for evaluation:
-	Reconstruction model: rendering samples per ray.
-	Hardware: GPU types, number of GPUs.
-	Time costs: training hours.
2.	There are some missing works in Section 2. Specifically, many works have attempted to incorporate 2D diffusion priors with NeRF or other 3D reconstruction model, like 3D Gaussian Splatting recently, for single-view or sparse-view reconstruction. The paper only provides two outdated methods, i.e. DreamFusion and Zero-1-to-3, and claims that these models do not consider the uncertainty. However, there are also many works that utilize uncertainty measure during reconstruction to improve view consistency, e.g. [1-4]. Specifically, [4] also uses diffusion as priors, contains uncertainty measures, and targets sparse-view reconstruction, which is very similar to the claimed contribution. Although they haven’t released their code, the author(s) should also consider mentioning these works in the related work section.
3.	The motivation is not clear enough. The paper mentions that 3D reconstruction problem is usually approached with deterministic gradient based optimization methods, while 3D generation is typically addressed with probabilistic models, so the author(s) propose to utilize posterior sampling for probabilistic reconstruction. However, they do not mention the pros and cons of probabilistic methods. In specific, if current 3D reconstruction is good enough, why do we need non-deterministic approaches and how can we benefit from it?
4.	Insufficient comparison with baselines. Only Figure 7 and Table 2 show the comparison with previous works, where Figure 7 has only one example and one baseline, and Table 2 has two baselines. To help evaluation and prevent cherry pick, the author(s) may consider showing more examples from different views/subjects. Also, as mentioned, many 3D works with 2D diffusion can achieve single-view or two-view reconstruction, for example, SDS-based works, i.e. DreamFusion, DreamGaussian. Two baselines may lead to difficulties in evaluation.


[1] L Goli et al., Bayes' Rays: Uncertainty Quantification for Neural Radiance Fields, CVPR 2024.

[2] J Shen et al., Conditional-flow nerf: Accurate 3d modelling with reliable uncertainty quantification, ECCV 2022.

[3] N Sünderhauf et al., Density-aware nerf ensembles: Quantifying predictive uncertainty in neural radiance fields, ICRA 2023.

[4] X Liu et al., Deceptive-NeRF/3DGS: Diffusion-Generated Pseudo-Observations for High-Quality Sparse-View Reconstruction, ECCV 2024.

**Questions:**

1.	The paper claims it as an efficient and accurate 3D reconstruction from observations, but did not provide comparison of time/computational/hardware complexity. Specifically, how would you compare this efficiency, and could you provide a quantitative comparison?
2.	See weakness.

---

> ### Comment · Reviewer_AMeD · 2024-11-26
>
> After reading the reviews from other reviewers, I am certain that a number of concerns are not properly resolved in this submission. Specifically, except the discussion about the novelty, which cannot be changed in the discussion period, I conclude that there are three major concerns:
>
> -  **Confusion on method.**
> Initially, I was impressed by the results of uncertainty in Fig. 1. However, reviewer o44T's concern regarding how to calculate the uncertainty has not been properly addressed.
>
> - **Missing prior work.**
> As stated in the second weakness of my official review and in other reviewers' comments, prior works are missing in the submission.
>
> - **Insufficient experiment.**
> The experimental results are not sufficient in terms of the number of examples and the difficulty.
>
> As the deadline of public discussion is approaching, I decide to temporarily adjust my suggestion score, and I'm hopeful that these issues can be further addressed in the revised version.

---

> > ### Author Response · Authors · 2024-11-26
> >
> > We thank the reviewer for their work and positive feedback. We give a general response addressing the main issues raised by all reviewers in a separate post, and here we address all issues raised by reviewer AMeD. We apologize for the delay in the response and hope that by addressing the issues and given the remaining discussion period until Dec 2, the reviewer could raise their score back to its original value.
> >
> >
> > ### **New concern regarding uncertainty computation**
> > As we answer reviewer o44T, the uncertainty maps in our method show the variance between the rendered images obtained from multiple posterior samples. Specifically, when the model is guided by an observation, such as an image of the top half of a chair, the sampled latents will consistently reconstruct the observed top half while producing diverse completions for the unobserved bottom half. This variability translates into high variance in the bottom half, which is visualized as uncertainty. The process is task-agnostic, as it directly reflects the variability of the posterior conditioned on the given observations.
> >
> >
> > ### **W1/Q1 Implementation details and efficiency**
> > Implementation details (to be added to the new manuscript):
> > - Reconstruction Model:
> > Rendering Samples per Ray: 220 samples per ray.
> > Training Configuration: 4096 rays collected from all training images in the scene per step, with 2 scenes per batch.
> > - Hardware:
> > GPU: 1 NVIDIA GeForce RTX 4090.
> > - Time Costs:
> > Training Duration: Approximately 1 day.
> >
> > Regarding efficiency, we mean in terms of amount of conditioning information (e.g. sparse pixels). We will make this clearer in the revised paper.
> >
> > ### **W2 Missing Prior Work**
> > We thank the reviewer for the relevant prior work that we missed, and we will mention all of them in the revised paper. As we also mention in the general comment, prior work related to uncertainty computation [1,2,3] differs from ours fundamentally because their uncertainty is based on detecting inconsistencies in the reconstructions of a single scene, highlighting areas which are hard to reconstruct. In contrast, our method relies on a prior that provides information about how these areas could be reconstructed.  The work in [4] also uses a prior to compute uncertainties, however it is a 2D prior that is used to “fix”  images generated by NeRF (therefore the prior does not contain 3D information by itself). We also address more prior work in the general response.
> >
> >
> > ### **W3 Motivation**
> > While traditional 3D reconstruction methods perform well when sufficient information about the scene is available, they struggle with handling occlusions and missing data. Probabilistic methods, on the other hand, provide a distinct advantage by explicitly modeling uncertainty. In such approaches, the observed parts of the scene remain consistent across multiple samples, while unobserved or occluded parts result in diverse plausible completions. This not only enables the generation of reasonable reconstructions in the face of missing information but also provides valuable insights into the certainty of the reconstructed scene, which is not achievable with deterministic methods.
> >
> > ### **W4 More comparisons**
> > Most of the results we demonstrate are for tasks that to the best of our knowledge were not shown to be possible by any other model before (e.g. Figure 1 and 8). Prior work mostly focuses on reconstruction from a single view which is only one out of many tasks we consider. The purpose of Figure 7 and Table 2 was mainly to show that methods that are optimized on pointwise metrics fail to capture the full distribution of possible reconstructions.

---

> > > ### Comment · Reviewer_AMeD · 2024-11-27
> > >
> > > **The description of how to compute the uncertainty is still unclear..**
> > >  Do you mean you calculate the uncertainty during inference for one example? Or you generate 10 examples with the identical input condition, and calculate the variance between these 10 examples? If you use the second approach, which computes the variance across several times of inference results, I am certain that this can be done in most of the prior works.
> > >
> > > **Implementation details.**
> > > This concern has been addressed. Please add these details to the revised version.
> > >
> > > **Prior works.**
> > > See my comments on the general response for details.
> > >
> > > **Motivation.**
> > > The motivation is interesting to me. This concern has been addressed.
> > >
> > > **More comparisons.**
> > > As mentioned by other reviewers, this paper needs a more complicated dataset, as well as more baselines and quantitative results, to demonstrate the benefits of the proposed approach.

---

> > > > ### Author Response · Authors · 2024-12-01
> > > > **uncertainty computation**
> > > >
> > > > We have added a description of the uncertainty computation (lines 481-482). It is the second approach you described. Given the observation, we generate multiple samples of the same scene, for each sample we render an image from the same view and compute the variance of the pixel values. This is a simple way to demonstrate the remaining uncertainty after combining the prior information (e.g. this scene should include a type of chair) and the observation (e.g. the values of certain pixels from certain views).

---

> ### Comment · Reviewer_AMeD · 2024-12-02
>
> Thanks for the updates. My questions have been properly answered after the discussion. As a conclusion, I believe this paper uncovers a simple yet effective approach to connect diffusion models with 3D reconstruction models, where it enables training a diffusion model in 3D by pretraining a conditional radiance field with corresponding latent space. However, there are still some missing experiments, (e.g. more complicated datasets, quantitative evaluation, more baselines), which affects the fairness of evaluation. Consequently, I decided to maintain my score at 6.

---

### Official Review · Reviewer_DFAS · 2024-11-02

**Soundness:** 2
**Presentation:** 2
**Contribution:** 2
**Rating:** 3
**Confidence:** 5

**Summary:**

The paper combines a neural filed representation of 3D objects with a latent diffusion model for 3D novel view synthesis and completion from partial visible information.
In particular, it presents a method that learns prior distribution of 3D objects using a conditional neural field and a latent representation.
Then, a latent diffusion model is introduced to learn the prior over the proposed presentation.
Experiments are conducted on simple synthetic SRN cars and Objaverse-lvis chair, and demonstrate reasonable visual results on both novel view synthesis and completion,
outperforming the pixeNeRF baseline, which is a very old approach.

**Strengths:**

### S1 --- A good attempt at an interesting and valuable problem

- The task of novel view synthesis from single image is interesting, especially from partially visible inputs. This is very valuable given the heavy occlusion in the real applications, and has been attracting growing interest in the community.
- This paper makes a good initial attempt to tackle this problem in very simple synthetic cases (synthetic objects with single category, rendered with multiple views). The method is reasonable to combine the CNF and diffusion model.

**Weaknesses:**

### W1 --- Significance is not well demonstrated
- The proposed idea is only a very specific, minor change in SSDNeRF  --- basically using a slightly different conditional neural field (CNF) to replace the original NeRF in SSDNeRF, while the rendering is still the volume rendering. Fundamentally, I am not fully convinced that it is even crucial to use this claimed new representation. In theory, the SSDNeRF also used the tri-planes representation for xy, yz, and xz, and then do the rendering.
- While the authors learn the latent representation for each object first, this small change is not so significant.
- In general, I do not think the paper have demonstrated the significance of the proposed change clearly enough. The baseline SSDNeRF model seems to do quite well on these datasets already.  The experimental results do not demonstrate the significance of the proposed methods. Besides, the current interesting towards more on the open-world category. The paper would be stronger to try some more challenging datasets.

### W2 --- Confusion on method
- How many latent vectors for each dataset? If we need to define a latent vector for each instance, it will be very expensive to learn this prior distribution.
- How could we enforce each latent vector corresponding to one instance? If they are paired, how do we match the new instance to the latent space? In L318-L320, the author claimed "...test scenes are used to optimize the scene latents while freezing the model's weights". In this way, how many steps we need to do for the optimisation? And how expensive of this optimisation step?
- Generally, $z_{t-1}$ still has a large gap to the clean latent. How could this be used for the Reconstruction model for optimisation?

### W3 --- Experiment setting is too simple
- The current experiments are conducted on very simple synthetic data with one special category, which has almost been addressed in the past two years.
- While the authors introduce a novel and interesting setting with partially visible information, the pixelNeRF is a too old baseline, which is not good enough to support the importance of the proposed method.

### W4 --- Missing prior work
- First, the decoder-only GaussianCube has been used to represent open-world category and is combined with the 3D diffusion for 3D generation and conditional generation.
- VQ3D is also another latent representation for the in the wild 3D objects.
- More feed-forward single-view 3D object reconstruction and novel view synthesis should be discussed, such as 3DIM.

**Questions:**

- What's the key difference between the proposed method to the existing SSDNeRF? A deeper discussion is necessary to highlight the main contribution of this manuscript.
- The proposed pipeline works only on one special category, which is very limited. For example, the similar GaussianCube also leans a decoder-only representation, but with the latest 3D Gaussian representation, and then use a 3D U-Net to deal with the generation and conditional generation. However, they verify the idea on various categories. Why this model can only work on a special category?
- The baseline model PixelNeRF is too old, which is hard to be considered as a baseline to demonstrate the effective of the proposed method. The latest 3DiM, zero-1-to-3, One-2345, Free3D, SV3D and others should be discussed and one of them can be used for the latest baseline.

---

> ### Author Response · Authors · 2024-11-26
>
> We thank the reviewer for their work and constructive comments. We give a general response to the main issues raised by all reviewers in a separate post, and address all the issues raised by  reviewer DFAS here.
>
> ### **W1/Q1 Significance is not well demonstrated**
> The main novelty in our work (specifically compared to SSDNeRF) is in the way the model is used at test time to generate multiple posterior samples rather than optimizing pointwise metrics. Figure 7 and Table 2 show that this can be contradictory - SSDNeRF performs better on the metrics but fails to cover the full posterior distribution of possible reconstructions.
>
>
> ### **W2 Confusion on method**
> We will make these details clearer in the revised paper.
> Dataset size = number of latent vectors. A latent representing each scene.
> There are N scenes / N latents. Each scene’s training images are used to optimize its own latent only.
> At training time, the whole model is optimised, including the latents and the decoders.
> At test time, a new latent (initialised with zeros) is coupled to the new scene and optimised using the learned (now frozen) decoders.
> The process of posterior sampling  involves gradual guidance. The denoising step focuses on generating a plausible latent representation, while the reconstruction step guides the denoised latent toward the "clean latent" that aligns with the given views. This iterative interaction ensures that the latent becomes both plausible and consistent with the observed data over successive steps.
> We tested our method across multiple categories with a small number of scenes in each, and the model successfully learned to handle multiple categories. However, as the number of scenes increases—whether within a single category or across multiple categories—the number of latents to optimize also grows. This growth introduces a practical limitation due to computational resource constraints.
>
> ### **W3/Q2 Experiment setting is too simple**
> Due to limited access to computational resources, we cannot run experiments on large scale datasets. We therefore focus on single category data, which we believe is enough to demonstrate the effectiveness of our method, and specifically its ability to incorporate prior knowledge into the 3D reconstruction pipeline. We demonstrate capabilities that involve high levels of uncertainty and to the best of our knowledge these tasks were not shown to be possible before, including not with modern methods. The goal of the comparisons to PixelNeRF and SSDNeRF in Figure 7 and Table 2 is to show on one hand that our results are at the level of SOTA methods (where SSDNeRF is our representative SOTA baseline), and on the other hand that optimizing pointwise metrics do not necessarily lead to a faithful posterior distribution of possible reconstructions (e.g. the different possible car colors).  Regarding GaussianCube, please see next comment.
>
> ### **W4/Q3 Prior Work**
> Regarding PixelNeRF, please see previous comment.
> We thank the reviewer for providing more relevant prior work that we missed. We will mention all these works in the related work section of the revised paper. As we also elaborate in the general comment posted separately, most of these works target the “graphics” problem of generating high quality 3D structures usually conditioned on an input image. Our aim is related to the “perception” problem involving a faithful prediction of the full posterior over 3D reconstructions. When we compare our method to SOTA for reconstruction from single image we chose SSDNeRF as the SOTA representative as it is closest to our model and enables a fair comparison on SRN cars. Regarding GaussianCube, this model shows promising results using a generative model over a Gaussian Splatting representation. Our contribution is orthogonal to theirs. We believe that applying our method on top of their model is a good future direction to explore.

---

### Official Review · Reviewer_o44T · 2024-11-03

**Soundness:** 3
**Presentation:** 3
**Contribution:** 3
**Rating:** 6
**Confidence:** 3

**Summary:**

This paper presents an approach to 3D scene reconstruction by combining neural radiance fields (NeRF) with probabilistic diffusion models. The method uses a two-stage training process: first, it trains a reconstruction model that compresses 3D scenes into compact latent representations using a tri-plane structure, and second, it trains a diffusion model as a prior over these latent representations. During inference, the method performs posterior sampling using the trained diffusion model guided by reconstruction error, allowing it to generate multiple plausible 3D reconstructions that are consistent with the input observations. The authors demonstrate that their method achieves competitive results with existing approaches while showcasing the model's additional capability of inference on a wide variety of tasks with different levels of uncertainty.

**Strengths:**

1. The authors have provided a well-rounded algorithm that combines CNF with compressed latent, Diffusion prior training for the distribution of latent, and a strategy that combines the prior and the rendering algorithm for sampling from the posterior given the observations. The algorithm is mostly sound and supported by theoretical basis from the Langevin sampling process, and empirical evidence also illustrates its competitiveness with similar methods.
2. The ability to perform inference on 3d reconstruction tasks using widely different observations appears to be novel and is quite interesting. Empirical evidence also shows that the influence of the prior is in proportion to the uncertainty of the 3d structure given the observations. The flexibility and effectiveness of such a posterior sampling strategy are valuable and can be expanded upon.
3. The two training steps and the sampling strategy are well-explained in the methods section, with clear graphic illustrations and experiment results to support the incentive for the algorithm. The paper is overall easy to understand.

**Weaknesses:**

1. In the contributions section the authors mentioned "We show that considering the full posterior can lead to better reconstruction and provide additional insight such as 3D uncertainty maps." However, I do not see a clear description of how the 3D uncertainty maps are calculated given the overall pipeline.
2. Although there is plenty of qualitative evidence to support the method, quantitative analysis is limited. It is also worth noting that the Table 2 results show the method does not provide the same accuracy compared to its peers. The paper could benefit from a more comprehensive summary of quantitative results.

**Questions:**

1. line 478-479 notes that "and an uncertainty map computed by the variance of ten generated samples". This is the only comment to how uncertainty can be calculated in the method. Given uncertainty is an important factor of the work, is it possible to give a clearer explanation on how uncertainty is retrieved? Does the uncertainty measure depend on the task or is it task agnostic?

2. Why are there only limited quantitative results? Have comparisons been done with other methods beyond 1/2 view reconstruction or latent reconstruction? (i.e. reconstruction with noisy/partial images, sparse points/depth)

3. In the last step of the posterior sampling, the gradient is used to update $z_{t-1}$ and is applied directly. Is a step size/ weight here that can be used here to control the influence of the observation?

---

> ### Author Response · Authors · 2024-11-26
>
> We thank the reviewer for their work and positive feedback. We address the main issues raised by all reviewers in a general response posted separately, and answer all issues raised by reviewer o44T here. We hope that addressing these issues will lead to the reviewer raising their score.
>
>
> ### **W1/Q1 Uncertainty maps:**
> The uncertainty maps in our method show the variance between the rendered images obtained from multiple posterior samples. Specifically, when the model is guided by an observation, such as an image of the top half of a chair, the sampled latents will consistently reconstruct the observed top half while producing diverse completions for the unobserved bottom half. This variability translates into high variance in the bottom half, which is visualized as uncertainty. The process is task-agnostic, as it directly reflects the variability of the posterior conditioned on the given observations.
>
> ### **W2/Q2 More quantitative results:**
> We were unable to run SSDNeRF on Objaverse Chairs to produce more comparable quantitative results. Comparisons on SRN cars in  Figure 7 and Table 2 show that while SSDNeRF performs better on pointwise metrics it fails to cover the full posterior distribution of possible reconstructions.  Due to limited access to computational resources, we cannot run experiments on many different datasets. We therefore focus on two experiments from two datasets which we believe demonstrate the effectiveness of our method, and specifically its ability to incorporate prior knowledge into the 3D reconstruction pipeline.
>
> Given that the goal of our work is to enable 3D reconstruction which is faithful to the uncertainty of the underlying scene, it is not straightforward to develop a quantitative evaluation method (we discuss this further in the general response). We therefore focus on qualitative results demonstrating the consistency and diversity of the generated samples for different levels of inherent uncertainty (Figures 1 and 6).  We also compare qualitative results of reconstruction from noisy observations compared to TensoRF in Figure 8.
>
>
> ### **Q3 step size:**
> Yes, the step size (denoted as s) in the gradient update for Z_{t-1}​ is used to control the influence of the observations during posterior sampling. We will clarify this point in the revised manuscript to ensure its role and impact are explicitly communicated.

---

### Official Review · Reviewer_uFkz · 2024-11-04

**Soundness:** 2
**Presentation:** 2
**Contribution:** 1
**Rating:** 3
**Confidence:** 4

**Summary:**

This paper proposes to combine diffusion prior and conditional gradient from reconstruction model to achieve posterior sampling. The authors choose to use latent code to representation each scenes and learn an mapping from latent code to tri-plane representations by minimizing the image loss after rendering. They try to prove the effectiveness of their method by showing reconstruction quality.

**Strengths:**

1. They propose to use latents to represent various objects in a dataset, and train diffusion on latents, which is reasonable and efficient to approximate the distribution of the dataset.
2. They provide the possibility of generating 3D uncertainty maps after training a "generative prior".
3. They provide the results of extensive experiments like noisy observed images, sparse observed images reconstruction, and they get relatively better results.

**Weaknesses:**

1. The method lacks more novelty, which has been proposed really similarly in previous works.
2. The tasks are relatively easy and the results are few and not impressive.

**Questions:**

1. Diffusion models are very likely to memorize rather than generalize when the data scale is relatively small, which has been studied and proved by many works. So can the authors explain, whether the results of your posterior sampling are generalizable or not, as the data amount is small? Also, the test dataset size is too small to provide more solid proofs and insights.
2. Basically, given 100 images about a scene, we can directly get a NeRF. Can the author explain what is the need for optimizing a latent? Or can the author prove that the NeRF generated directly is worse that the results of your method?
3. Also, we can train the diffusion model directly by CFG(classifier-free-guidance), which also play the role of posterior sampling. What is the advantage of using RM gradient, which is also similar with conditional score in CFG?

---

> ### Author Response · Authors · 2024-11-26
>
> We thank the reviewer for their work and constructive comments. We address the main issues raised by all reviewers in a general response posted separately, and answer all issues raised by reviewer uFkz here. We hope that addressing these issues will lead to the reviewer raising their score.
>
>
> ### **W1 Novelty:**
> The main novelty in our work (specifically compared to SSDNeRF) is in the way the model is used at test time to generate multiple posterior samples rather than optimizing pointwise metrics. Figure 7 and Table 2 show that this can be contradictory - SSDNeRF performs better on the metrics but does not cover the full posterior distribution of possible reconstructions.
>
> ### **W2 More experiments:**
> Due to limited access to computational resources we evaluate our method on relatively small datasets however we demonstrate capabilities that are not easy as they involve high levels of uncertainty. To the best of our knowledge these tasks were not shown to be possible before.
>
> ### **Q1 Memorization:**
> Figures 1 and 6 demonstrate diverse scene completions when provided with a partially observed scene, suggesting that the diffusion model has not overfit to the dataset. We attribute this to the compact latent representation employed during training, which reduces the risk of overfitting and allows effective training even with a relatively small dataset.
>
> ### **Q2 Full NeRF:**
> While it is true that a NeRF can be directly generated from 100 images, our method is designed to address scenarios where a compact and efficient representation is critical. By optimizing a latent representation, we enable the integration of a diffusion prior in the second stage. This approach facilitates posterior sampling, which is the core contribution of our work. In posterior sampling, only a single image—or even a small part of an image—is utilized to produce a latent representation of the scene. This latent captures the essence of the partially observed scene in a more structured and efficient manner than attempting to directly model the 100 images themselves.
>
> ### **Q3 CFG:**
> Our approach is using a guidance-based approach to generate posterior samples. The guidance is computed using the gradients of a likelihood term which is formed from the reconstruction model. In contrast to standard CFG, this means that we do not need to train a conditional diffusion model in advance, and that we can use the same model to solve lot’s of different tasks. All our experiments are using the same trained model, while using standard CFG would require training a different conditional model for each task (e.g. conditioning on single images, sparse pixels, depth information etc.)

---

### Author Response · Authors · 2024-11-26
**General Response**

We thank the reviewers for their work and for their constructive comments.
We give a general response to the main issues here, and in addition respond to each reviewer individually.

While the reviewers agreed that our work proposes a good method to tackle an interesting problem, and that the paper is well presented, some reviews were concerned about the novelty of our method, particularly compared to SSDNeRF. Additionally some reviewers asked for experiments on more datasets, more evaluations and stronger connection to relevant prior work. We address these issues here.


### **Novelty, Motivation and Significance**

The novelty of our work compared to SSDNeRF is mainly in the way it is used at test time, rather than in the model itself and the training method. We develop a method that treats the model as a prior of 3D scenes and enables generating posterior samples by combining a probabilistic model with gradient based optimization. Our main goal is to enable a 3D reconstruction method that faithfully reflects the true level of uncertainty about the underlying 3D scene. Rather than generating a single plausible high-quality example, we focus on reconciling prior knowledge with observations, maintaining the true uncertainty about the scene.

Figure 7 in our paper shows that SSDNeRF fails to capture the true distribution of the scene under uncertainty. This is because even though it uses a similar generative model to ours, it is not treated as a probabilistic prior, and it is optimized with respect to pointwise metrics which favor an average prediction rather than covering the diverse possible reconstructions.
We show that using posterior samples can lead to both efficient high quality reconstruction when enough information is given, and at the same time cover the full distribution of possible reconstructions.

Another way to describe the goal of our method vs. prior work is that prior work focuses mainly on **graphics applications** and we focus on a **perception application**, especially when prior knowledge about the scene is given. A real world example for such an application is 3D reconstruction in medical imaging, where understanding the full  spectrum of possible reconstructions is more important than getting very sharp predictions.

We will emphasize these points further in the paper.

---

> ### Author Response · Authors · 2024-11-26
>
> ### **More datasets, evaluations and prior work**
>
> Due to limited access to computational resources, we cannot run experiments on large scale datasets. We therefore focus on single category data, which we believe is enough to demonstrate the effectiveness of our method, and specifically its ability to incorporate prior knowledge into the 3D reconstruction pipeline.
> We note that using generative 3D models on real world datasets is still considered a very challenging problem even for recent high capacity models, and therefore most prior work still focuses on experimentation with synthetic data (e.g.  DreamFusion, zero-1-2-3, GaussianCube).
>
> Given that the goal of our work is to enable 3D reconstruction which is faithful to the uncertainty of the underlying scene, it is not straightforward to develop a quantitative evaluation method. One natural metric to use is log-likelihood, however this cannot be computed in our model since we can only generate samples (which need to go through a forward model therefore preventing log-likelihood computation based on probability flow).  We therefore focus on qualitative results demonstrating the consistency and diversity of the generated samples for different levels of inherent uncertainty (Figures 1 and 6).
>
> While we demonstrate our method on various different 3D reconstruction tasks (Figure 1, 6, 8), we also show that we are comparable to SOTA methods for the specific task of reconstruction from 1 or 2 views in Figure 7 and Table 2. However the main goal of this is to show the failure of methods that optimize metrics like PSNR and SSIM (e.g. SSDNeRF) to capture the diversity of possible reconstructions.  Since we could not train SSDNeRF on the chair dataset that we used (their provided training code does not converge to a working model), we had to perform this comparison on the same data (SRN cars) and evaluation protocol that was used in their paper. We emphasize that reconstruction from 1 or 2 images is only one task out of many tasks that we demonstrate in the paper. Moreover, we use the same model for all the conditional reconstruction tasks and unconditional sampling, and in contrast, prior work usually focuses on one task (for example SSDNeRF uses different models for conditional and unconditional generation of scenes).
>
> We thank the reviewers for providing additional relevant prior work that we missed. We will mention them in the paper, and emphasize their relation to our work. Prior work related to uncertainty computation [1,2,3] differs from ours fundamentally because their uncertainty is based on detecting inconsistencies in the reconstructions of a single scene, highlighting areas which are hard to reconstruct. In contrast, our method relies on a prior that provides information about how these areas could be reconstructed.  The work in [4] also uses a prior to compute uncertainties, however it is a 2D prior that is used to “fix”  images generated by NeRF (therefore the prior does not contain 3D information by itself). 3DIM [5] uses a novel view image prior to generate samples for training NeRF. In contrast to our method the prior does not contain explicit 3D grounding. VQ3D [6] grounds the novel view generation on a triplane representation like ours but focuses on generating imagenet  images rather than probabilistic perception tasks like we do.  As for the recent work of GaussianCube [7] which shows very good results in 3D generation, we think that combining our posterior sampling methods on top of their model could be interesting future work.
>
>
> [1] L Goli et al., Bayes' Rays: Uncertainty Quantification for Neural Radiance Fields, CVPR 2024.
>
> [2] J Shen et al., Conditional-flow nerf: Accurate 3d modelling with reliable uncertainty quantification, ECCV 2022.
>
> [3] N Sünderhauf et al., Density-aware nerf ensembles: Quantifying predictive uncertainty in neural radiance fields, ICRA 2023.
>
> [4] X Liu et al., Deceptive-NeRF/3DGS: Diffusion-Generated Pseudo-Observations for High-Quality Sparse-View Reconstruction, ECCV 2024.
>
> [5] D Watson et al., 3DIM: Novel View Synthesis with Diffusion Models
>
> [6] K Sargent et al., VQ3D: Learning a 3D-Aware Generative Model on ImageNet
>
> [7] B Zhang et al., GaussianCube: A Structured and Explicit Radiance Representation for 3D Generative Modeling

---

> ### Comment · Reviewer_AMeD · 2024-11-27
> **A few comments on the general response**
>
> **Evaluation**
> Although the author(s) claim that log-likelihood cannot be computed in their model, I am certain that a number of metrics are still feasible and beneficial for the evaluation. As the paper focuses on the probabilistic model to reflect the uncertainty of the underlying scene, for the "unseen" regions, it might be helpful if the author can measure the distance between the generated results and the ground-truth distribution. In this case, FID can be used (just like how it is used in SSDNeRF). Besides, since the author(s) mention that the experiments aim to demonstrate the diversity of generated result, the author(s) should also consider to measure this diversity. Similar to the tasks of conditional image generation, precision and recall can be useful in this case.
>
> **Relation to prior works**
> It is crucial not to overclaim the difference between your paper and the prior works. For example, it is debatable whether diffusion models trained on 2D images have 3D understanding. Previous works often incorporate 2D priors to enhance 3D reconstruction, especially for single view or sparse view reconstruction. These kinds of methods can basically be divide into two categories:
> 1. SDS-based approaches (e.g. DreamFusion, ProlificDreamer), which aims to distill the knowledge from 2D diffusion models to 3D models by aligning the distribution from random poses.
> 2. Data-based approaches (e.g. Deceptive-NeRF/3DGS, SyncDreamer, InceptionHuman), which aims to design a pipeline to generate consistent multi-view images from pretrained 2D diffusion models.
>
> These approaches both show that 2D diffusion like Stable Diffusion or Zero-1-to-3 has 3D understanding to some degree. Your contribution, to my knowledge, is to backpropagate or distill the 3D understanding from the shared CNF to the diffusion model at training time, and further enable uncertainty-aware prediction at testing time. At the same time, your design avoids the complicated training algorithm in SSDNeRF. However, it is still good if you can demonstrate how the 3D-aware diffusion distinguish your work from other methods based on 2D diffusion models. To this extent, you would need more comparison with 2D-based approaches, qualitatively and quantitatively.

---

> > ### Author Response · Authors · 2024-12-01
> >
> > We thank reviewer AMeD for their engagement in the discussion.
> >
> > Regarding evaluation, we agree that some quantitative metric for diversity and faithfulness could be devised. However, these metrics would be ad-hoc and not comparable to standard quantitative benchmarks used to evaluate previous approaches. Therefore, we believe that demonstrating the qualitative capacities of our method is more informative.
> >
> > Regarding prior work on 2D diffusion, we agree with your comment that 2D models can sometimes have a high level of 3D understanding. The main advantage in our method is that having a model directly on a 3D representation unlocks more diverse inference tasks that are not necessarily based on observations that are downstream from the 2D images (for example using depth information). We have updated the related work section based on your comment.

---

### Meta-Review · Area_Chair_Df7F · 2024-12-14

**Metareview:**

The authors propose a method for posterior sampling in a 3D latent space to perform reconstruction from single observations. The work received negatively leaning reviews, criticizing the incremental nature, especially with respect to SSDNeRF, the small scale experimental setup on only two small, synthetic datasets and lacking comparisons to newer baselines.
I agree with these concerns and recommend to reject.

While I appreciate the goal of modeling the full range of inherent uncertainty in the reconstruction task and consider it an important research problem, I agree with the reviewers that the success of this method has not been successfully demonstrated in the experiments.

**Additional Comments On Reviewer Discussion:**

The reviewers clearly stated their concerns in the first round. However, most concerns were not adequately addressed by the authors. The experimental setup was not extended and no further baselines have been added.  While the authors were able to clarify their contribution with respect to previous works, the reviewers remained concerned about the validation of effectiveness.

---

### Decision · Program_Chairs · 2025-01-22

Reject